# Tumor- and Fibroblast-Derived Cell-Free DNAs Differently Affect the Progression of B16 Melanoma In Vitro and In Vivo

**DOI:** 10.3390/ijms25105304

**Published:** 2024-05-13

**Authors:** Alina A. Filatova, Ludmila A. Alekseeva, Aleksandra V. Sen’kova, Innokenty A. Savin, Khetam Sounbuli, Marina A. Zenkova, Nadezhda L. Mironova

**Affiliations:** 1Institute of Chemical Biology and Fundamental Medicine, Siberian Branch of the Russian Academy of Sciences (SB RAS), Lavrentiev Ave., 8, Novosibirsk 630090, Russia; a.filatova2@g.nsu.ru (A.A.F.); alekseeva.mila.23@yandex.com (L.A.A.); senkova_av@niboch.nsc.ru (A.V.S.); savin_ia@niboch.nsc.ru (I.A.S.); khetam.sounbuli.edu@gmail.com (K.S.); marzen@niboch.nsc.ru (M.A.Z.); 2Faculty of Natural Sciences, Novosibirsk State University, Novosibirsk 630090, Russia

**Keywords:** cell-free DNA, metastasis, tumor progression, fibroblast, melanoma

## Abstract

It is widely postulated that the majority of pathologically elevated extracellular or cell-free DNA (cfDNA) in cancer originates from tumor cells; however, evidence has emerged regarding the significant contributions of other cells from the tumor microenvironment. Here, the effect of cfDNA originating from murine B16 melanoma cells and L929 fibroblasts on B16 cells was investigated. It was found that cfDNAL929 increased the viability and migration properties of B16 cells in vitro and their invasiveness in vivo. In contrast, cfDNAB16 exhibited a negative effect on B16 cells, reducing their viability and migration in vitro, which in vivo led to decreased tumor size and metastasis number. It was shown that cell treatment with both cfDNAs resulted in an increase in the expression of genes encoding DNases and the oncogenes *Braf*, *Kras*, and *Myc.* cfDNAL929-treated cells were shown to experience oxidative stress. Gene expression changes in the case of cfDNAB16 treatment are well correlated with the observed decrease in proliferation and migration of B16 cells. The obtained data may indicate the possible involvement of fibroblast DNA in the tumor microenvironment in tumor progression and, potentially, in the formation of new tumor foci due to the transformation of normal cells.

## 1. Introduction

The study of circulating cell-free DNA (cfDNA) has gained significant momentum in recent decades, with tens of thousands of cfDNA studies being published each year [1,2,3,4,5]. Most of these studies have focused on the applications of cfDNA in medicine, such as liquid biopsies, sources of prognostic biomarkers, and the development of cfDNA-based test systems, including noninvasive prenatal testing [3,5,6]. Another important area of research is devoted to the origin, composition, and effects of cfDNA on cells and organisms, as well as other related aspects [7,8].

cfDNA appears in the bloodstream through the release of genomic DNA due to cell destruction via apoptosis or necrosis [9]. In individuals without any health complications, most cfDNA is derived from hematopoietic cells [10]. However, in the case of injuries and inflammatory or oncological diseases, the resulting foci of inflammation and necrosis contribute to an increase in the concentration of cfDNA in the bloodstream and changes in its composition [10,11]. Thus, cfDNA carries genetic and epigenetic information from the cell and tissue from which it originated [12].

It has been established that fragments of tumor-specific DNA appear in cfDNA in the blood of patients with cancer [13]. In addition to tumor cells, cells from the tumor microenvironment and normal cells adjacent to tumor cells contribute to the composition of cfDNA [10]. Thus, tumors can be affected by cfDNA derived from cells in the tumor microenvironment, such as cancer-associated fibroblasts (CAFs), tumor-associated neutrophils (TANs), and tumor-associated macrophages (TAMs).

cfDNA originating from tumor cells is heterogeneous in terms of length and composition and contains mutant sequences of oncogenes, such as *Kras* and *P53*, as well as overrepresented sequences, such as fragments of oncogenes and mobile genetic elements (MGEs). In addition, it contains regions reflecting genome instability and displays changes in methylation patterns and oxidative modification of nucleic bases originating from cancer cells. Such cfDNA has the potential to induce various effects in non-neoplastic cells. TLR-9 and CCDC-25 receptors recognize DNA based on specific patterns of methylation and the presence of 8-oxoguanine in the DNA composition, respectively [14,15]. Activation of these receptors leads to the initiation of apoptotic response, systemic inflammation, cytokine secretion, and other related processes.

A few recent pioneering studies have investigated the effects of tumor-specific cfDNA on the development of pathological processes [16,17,18,19,20,21,22,23]. The ability of tumor-specific cfDNA to penetrate into normal cells, enter the nucleus, and integrate into the genome of the recipient cell, thereby affecting the expression of many genes and microRNAs associated with oncogenesis and cell migration, has been demonstrated [16,17,18,19]. A number of studies have shown that circulating cfDNA of tumor origin has the ability to enter normal cells, causing changes that lead to the formation of a tumor microenvironment, a premetastatic niche, and metastases [17,20]. Our recent studies have shown a correlation between a decrease in the concentration of tumor-specific cfDNA in the blood of tumor-bearing mice and a decrease in the size of the primary tumor/number of metastases, which indicates the functional significance of cfDNA for tumor progression [21,22,23].

It is believed that tumor-derived cfDNA circulating in the blood of patients with oncological diseases constitutes a relatively small part of the entire cfDNA, and the proportion of this “malignant” part reflects the severity of the disease. However, the functional role of cfDNA, including both the “malignant” and “healthy” components originating from tumor-associated fibroblasts, endothelial cells, and cells of the immune system, remains a subject of investigation. In this regard, studying the role of circulating “malignant” and conditionally “normal” cfDNAs in the initiation of new metastatic foci is extremely interesting.

In this study, we investigated the effect of cfDNA from the culture media of murine B16 melanoma cells and L929 fibroblasts on the invasive potential of B16 cells in vitro and in vivo and looked for correlations between the main parameters associated with tumor progression (tumor size and metastasis number) and other parameters, including DNase activity and cfDNA concentrations in blood serum and leukocyte numbers in the blood, spleen, and tumor. We found that cfDNA derived from tumor cells and fibroblasts differently affects the invasive potential of B16 melanoma in vitro and in vivo.

## 2. Results

### 2.1. Characterization of cfDNA from Conditioned Media of B16 and L929 Cells

Murine melanoma cells B16 were chosen as the source of tumor-specific cfDNA, and L929 fibroblasts were chosen as the source of “normal” cfDNA. cfDNA from conditioned media of B16 and L929 cells (hereinafter, cfDNA_B16_ and cfDNA_L929_, respectively) was isolated, and the abundance of oncogene and MGE fragments in the DNA samples was characterized by qPCR (Table 1). The genomic DNA of B16 and L929 cells (gDNA_B16_ and gDNA_L929_, respectively) was isolated and characterized in the same manner to determine whether the cfDNA profile was similar to the genomic DNA profile (Table 1).

gDNA_B16_ is characterized by a moderate abundance of tandem repeat fragments *B1_mus1*, *B1_mus2*, and *L1td1_orf2*, which encode transposition enzymes, and a low abundance of *L1_mus1* and *L1td1_orf1* fragments (Table 1). Among the oncogenes in gDNA_B16_, *Myc* fragments (all exons) were detected at a moderate level, and *Ras* fragments were detected at a low level. cfDNA_B16_ was characterized by a similar content of tandem repeat fragments and oncogenes, except for *L1td1_orf2* and *Ras* fragments, which were abundant at low and moderate levels, respectively (Table 1).

gDNA_L929_ is characterized by a low abundance of tandem repeats *B1_mus1*, *B1_mus2*, and *L1_mus1*, a moderate abundance of *L1td1_orf2*, and a negligible level of *L1td1_orf1*. Fragments of the oncogenes *Myc* and *Ras* were present in gDNA_L929_ at a low level (Table 1). In cfDNA_L929_, the abundance of *B1* and *Myc* fragments (exon 1 and exon 3) was moderate; the level of *Myc* fragments (exon 2) was at the same low level as in cfDNA_B16_, although it was sufficiently higher in gDNA_L929_; and *L1td1_orf1* fragments appeared at a low level (Table 1). *Hmga2* fragments were not detected in either genomic or cfDNA, regardless of their origin.

We tested the contribution of mitochondrial DNA to the composition of the isolated DNA. We did not find mitochondrial DNA sequences (*mt-Co3*, mitochondrial cytochrome c oxidase III, or *mt-Nd3*, mitochondrial NADH dehydrogenase 3) in any of the genomic DNA, and we also did not find them in cfDNA_L929_. In cfDNA_B16_, a number of *mt-Nd3* sequences were found, while *mt-Co3* was not detected. In addition, we checked the presence of oxidized DNA by measuring 8-oxoG in both types of cfDNA and in the conditioned medium using an ELISA kit, but we did not detect the presence of 8-oxoG in any sample.

### 2.2. Effect of cfDNA on B16 Cell Proliferation, Apoptosis, and Migration

The ability of cfDNA_B16_ and cfDNA_L929_ to induce apoptosis in B16 cells was investigated at two different cfDNA concentrations: 100 and 1000 ng/mL. cfDNA_B16_ at a concentration of 100 ng/mL did not induce apoptosis (*p* = 0.86), whereas cfDNA_B16_ at a concentration of 1000 ng/mL triggered apoptosis in 15–20% of cells for 24 h (*p* = 0.001) (Figure 1A, right panel). Under similar conditions, cfDNA_L929_ is not able to induce apoptosis in B16 cells (*p* = 0.72 and *p* = 0.79, respectively) (Figure 1A, middle panel, and Appendix A). cfDNA_B16_ in the concentration range of 100–1000 ng/mL decreased the viability of B16 cells up to 60% (*p* = 0.001 for a concentration of 1000 ng/mL) (Figure 1B). Interestingly, cfDNA_L929_ in the same concentration range slightly increased the viability of B16 cells (up to 20% compared to control cells); however, this difference was not statistically significant (*p* = 0.2) (Figure 1B).

The effect of cfDNA_B16_ on the motility of B16 cells was investigated using the scratch assay. The integrity of the monolayer formed by B16 cells was violated by scratching, and the rate of its filling the cells was monitored in the absence (control) and presence of cfDNAs at concentrations of 100 and 1000 ng/mL for 24 h (Figure 1C,D). In the control (intact B16 cells), the scratch was half filled with cells at the 24 h time point (Figure 1E, left panel). Both cfDNAs at a concentration of 100 ng/mL had a slight effect on the migratory activity of B16 cells (*p* = 0.89 and *p* = 0.88, respectively). B16 cells treated with cfDNA_B16_ at a higher concentration (1000 ng/mL) filled the scratch area by not more than 25% (*p* = 0.01) (Figure 1C–E). cfDNA_L929_ at the same concentration enhanced cell motility (*p* = 0.01) (Figure 1C). Because the scratch assay was performed in a serum-free medium, the likelihood that the increase in cell numbers could have arisen from the division of cells remaining in the scratch area rather than cell migration was extremely low. This is confirmed by the image of scratch healing observed in the presence of cfDNA_L929_, where it can be clearly seen that only a few cells remained in the lumen, and the movement of the cell front is distinctly visible (Figure 1E, upper middle panel).

To eliminate the effect of possible RNA contamination in DNA samples during isolation, cfDNA samples were treated with RNase A. It was found that such treatment (i) did not change the concentration of cfDNA and (ii) did not change the effects of cfDNA_B16_ on the viability and migration of B16 cells, indicating that phenol and column extraction methods allow for pure cfDNA samples lacking RNA contamination (Appendix A).

We also examined the presence of TLR9 receptors, known for their ability to sense CpG-DNA signals, in B16 melanoma cells. We tested the presence of TLR9 receptors on cells by co-staining with anti-TLR9 antibodies and propidium iodide (PI), which is known to not stain cells with intact membranes. As shown in Appendix A, 100% of B16 melanoma cells were stained with anti-TLR9 antibodies, regardless of PI staining. When cells were treated with a 0.5% Tween20 solution, which led to rupture of the cell membranes, it was shown that the number of cells stained with PI increased from 10% to 52%, whereas the intensity of TLR9 staining did not change. The data obtained allow us to suggest that B16 melanoma cells have TLR9 receptors on their cell membranes and are capable of receiving signals from DNA.

### 2.3. Effect of cfDNA on the Expression of Genes Encoding DNases, Oncogenes, and Genes Involved in Key Cell Processes

We analyzed the expression of genes that could potentially contribute to the observed effect of cfDNA: genes encoding DNases (*Dnase 1*, *Dnase 1l3*, *Dnase 1l1*, *Endog*, and *Dffb*) as participants in the response to exogenous DNAs; melanoma-related oncogenes (*Braf*, *Myc*, *Kras*, and *Raf1*) [24,25,26,27] and genes involved in driving melanoma progression (*Rac1* and *Rhoa*) [28]; apoptosis-related genes (*Tp53* and *Stat3*); respiratory chain genes (*Atp5f1b*, *Atp5If1*, *Coq10a*, *Cycs,* and *Cox6c*); cell division regulator genes (*Cdc42*, *Cdk4*, *Cdk6*, and *Ccnd1*); genes involved in the TLR9 pathway (*Hmbg1*) [29,30]; adhesion genes (*Icam*). Gene expression was analyzed in cells treated with cfDNA_B16_ and cfDNA_L929_. Because we expected that cell exposure to cfDNA would result in an increase in the expression of intracellular and secretory DNases, we first assessed the expression of genes encoding DNases. cfDNA caused a significant increase in the expression levels of *Dnase 1* and *Dnase 1l3* (secretory DNases), *Dnase 1l1* (intracellular DNase), as well as *Endog* (mitochondrial DNase) and *Dffb* (the apoptotic DNase) compared with untreated cells (Figure 2A). Among DNases, *Dnase 1* exhibited the most pronounced upregulation: its level increased 50-fold and 5-fold after treatment with cfDNA_L929_ and cfDNA_B16_, respectively. Meanwhile, the expression level of *Dnase 1l1* increased 5-fold regardless of the type of cfDNA used (Figure 2A). In addition, *Endog* expression increased by more than twice its baseline level in response to cfDNA_L929_ treatment.

In the control cells, no expression of *Dnase 1l3* and *Dffb* was observed; both cfDNAs stimulated the expression of *Dffb*, whereas the expression of *Dnase 1l3* was activated only by cfDNA_B16_ (Figure 2A). It is noteworthy that the expression of *Dnase 2a* (acidic lysosomal DNase) in the control cells was initially very high compared with other DNases, and its level dropped after treatment with cfDNA_B16_.

Specifically, we observed a substantial increase (ranging from 3.5- to 10-fold) in the expression levels of *Braf*, *Kras*, and *Myc* oncogenes upon cell treatment with both cfDNAs, and an increase in *Raf1* expression was observed exclusively in response to cfDNA_L929_ (Figure 2B). It is worth noting that cell exposure to cfDNA_L929_ significantly stimulated *Myc* expression more than cfDNA_B16_: 7-fold versus 3-fold for cfDNA_L929_ and cfDNA_B16_, respectively (Figure 2B). Furthermore, no notable changes in the expression levels of *Fos* and *Jun* genes were detected for either of the cfDNAs. It is important to mention that both *Fos* and *Jun* showed high expression levels in the control cells, which amounted to 5.2 and 4 relative units, respectively, when compared with the aforementioned oncogenes.

cfDNA_B16_ essentially decreased the expression level of *Hmgb1* driving TLR9 signaling, whereas cfDNA_L929_ had no effect. But it should be noted that the level of this gene was initially very high, and even after its decline remained at a fairly high level. Among genes involved in driving melanoma progression, *Rhoa* expression was reduced by a factor of three after treatment with both cfDNAs, whereas *Rac1* expression did not change (Figure 2C). Both types of cfDNA caused an increase in the expression levels of pro-apoptotic genes: a substantial increase in the expression of *Stat3* and a mild increase in the expression of *Trp53*. Among genes encoding components of the respiratory chain, *Atp5If1* and *Cycs* genes are of particular interest: *Atp5If1* exhibited a twofold increase in expression after treatment with cfDNA_L929_, whereas *Cycs* expression was reduced by a factor of three after treatment with both cfDNAs (Figure 2C). Similar to *Cycs*, downexpression trends of *Atp5f1b* were observed (Figure 2C). No significant changes in the expression levels of *Coq10a*, *Cox6c*, or *Atf5f1b* were observed in B16 cells treated with both cfDNAs (Figure 2C).

Changes were also noted in the expression levels of some genes that regulate the cell cycle. We found a tendency toward an elevation of *Cdk6* expression after treatment with cfDNA_B16_ and a 6-fold elevation of *Cdk6* expression after treatment with cfDNA_L929_ (Figure 2C). A similar dependence was observed for *Cdk4* expression; however, these data lacked statistical significance. No significant changes in the expression levels of *Cdc42* and *Ccnd1* were found (Figure 2C). Both cfDNAs had no effect on *Icam* expression in B16 cells (Figure 2C).

### 2.4. Effect of cfDNA on B16 Melanoma Development In Vivo

To assess the effect of cfDNA_B16_ on tumor invasive potential, two models of B16 melanoma were used. In the first model, B16 cells were implanted subcutaneously and formed a tumor node. In the second model, B16 cells were implanted intravenously and effectively generated metastases in the lungs without tumor node formation.

B16 cells were treated with cfDNA_B16_ or cfDNA_L929_ at a concentration of 100 ng/mL for 24 h, harvested, resuspended in saline buffer, and injected subcutaneously (0.1 mL) or intravenously (0.2 mL) into C57Bl mice. Six groups of mice were formed: 1, 2, and 3—intact B16 cells (control), B16 pretreated with cfDNA_L929_, and B16 pretreated with cfDNA_B16_, respectively, implanted s.c.; 4, 5, and 6—intact B16 cells (control), B16 pretreated with cfDNA_L929_, and B16 pretreated with cfDNA_B16_, respectively, implanted i.v. The scheme of the experiment is depicted in Figure 3A.

The obtained data showed that pretreatment of B16 melanoma cells with cfDNA_B16_ led to suppression of B16 growth, whereas cfDNA_L929_ intensified its development. In group 3, a 1.5-fold reduction in the primary size of tumor nodes compared with that in group 1 was observed (0.4 ± 0.06 vs. 0.58 ± 0.15 cm^3^, Figure 3B, Table 2). In group 2, the size of the tumor nodes was 1.5 times larger than that of the control (0.84 ± 0.06 vs. 0.58 ± 0.15 cm^3^, Figure 3B, Table 2). It is noteworthy that group 2 was characterized by a more pronounced weight loss in mice compared with the control group 1 (3.9 ± 1.9% vs. 2.7 ± 1.4%, Table 2), indicating a greater aggressiveness of the tumor progression process.

In the metastatic model, similar trends were observed. Group 6 was characterized by a 1.6-fold decrease in the number of metastases in the lungs compared with that in control group 4 (39 ± 6 vs. 56 ± 12 in group 4, Figure 3C). In group 5, the number of lung metastases was 1.2 times higher than that in group 4 (69 ± 9 vs. 56 ± 12, Figure 3C, Table 3). No effects on the body weight of mice in groups 5 and 6 were observed (Table 3).

The ratio of cfDNA concentration to DNase activity in the blood of mice with various tumors is an important characteristic of tumor progression [23]. In our experiments, we demonstrated that in control groups 1 and 4, the level of cfDNA increased by approximately two times compared with the healthy mice, whereas the level of DNase activity decreased by two times (Table 2 and Table 3). In the groups of mice (3 and 6) bearing B16 cells pretreated with cfDNA_B16_, the level of cfDNA decreased below that in the respective control groups (1 and 4), close to the level of the healthy animals. DNase activity increased to the level of the healthy animals (Table 2 and Table 3). In the groups of mice (2 and 5) bearing B16 cells pretreated with cfDNA_L929_, the cfDNA concentration did not differ from that of the respective controls, whereas the level of DNase activity decreased by two times relative to the controls (Table 2 and Table 3).

Histologically, the primary tumor nodes of B16 melanoma transplanted s.c. were represented by polymorphic or spindle-shaped atypical cells containing the brown pigment melanin (Figure 3D). Lung metastatic foci of B16 melanoma transplanted i.v. were located predominantly around the bronchi and blood vessels and had a structure similar to that of primary tumor nodes (Figure 3D). The number of mitoses in the tissue of primary tumor nodes in groups 1 and 3 was the same and amounted to approximately three in the testing area (Figure 3D, upper left and right panels, Table 2). In group 2, the numerical density of mitoses in primary tumor nodes was 1.8-fold higher than that in groups 1 and 3 (Figure 3D, upper panels, Table 2), indicating an aggravation of the proliferative activity of B16 melanoma after cfDNA_L929_ pretreatment. In the B16 metastatic model, the number of mitoses in lung metastatic foci in group 4 was the same as that in group 1 with primary tumor nodes (Figure 3D, left upper and lower panels, Table 2 and Table 3). In group 5, we observed a slight increase in the number of mitoses in lung metastases, whereas group 6 was characterized by a 5.8-fold reduction in mitosis numbers in metastatic foci compared with control group 4 (Figure 3D, middle and right lower panels, Table 3), demonstrating a dual effect of cfDNA of different origins on the proliferative potential of tumors.

Assessing peripheral blood parameters in mice with B16 melanoma transplanted s.c., in group 1, we detected a slight increase in the total leukocyte number and percentage of neutrophils in the peripheral blood of the mice compared with the healthy animals (Table 2). In group 3, we observed a slight decrease in the total leukocyte number in the blood, whereas the percentage of neutrophils decreased by 3.7 and 2.5 times compared with group 1 and the healthy mice, respectively (Table 2). Although the total number of leukocytes in group 2 was only slightly lower than that in group 1, the percentage of neutrophils was 2.3 times higher. A different picture was observed for B16 melanoma transplanted i.v. (groups 4, 5, and 6). The total number of leukocytes in the blood did not differ between the groups and was the same as that in the healthy mice, whereas the percentage of neutrophils in the blood increased in all groups and was 4.5–5 times higher than that in the healthy mice (Table 3). There were no statistically significant changes in the number of neutrophils in the spleen in all groups compared with that in the healthy animals (Table 2 and Table 3). Nevertheless, groups 2 and 5 bearing B16 cells pretreated with cfDNA_L929_ were characterized by a tendency toward a decline in neutrophil numbers in the spleen compared with groups 1 and 4, respectively (Table 2 and Table 3).

Afterward, it seemed interesting to study the cell composition of the infiltrated cells in primary tumor nodes in groups 1–3. Cell infiltration was predominantly represented by macrophages, lymphocytes, and neutrophils and was located at the border of unaltered tumor tissue and necrotic decay in the central part of the tumor. The representation of neutrophils in the cell infiltration in tumor nodes was infrequent and amounted to approximately 6% in the tumors of groups 1 and 3. In group 2, we observed a slight increase in the ratio of neutrophils in the entire cellular infiltration of the tumor up to 8.6% (for clarity, see Figure 3E).

Thus, the growth of B16 melanoma pretreated with cfDNA_L929_ affects neutrophil representation in tissues and biological fluids and is accompanied by an increase in the number of neutrophils in the peripheral blood and primary tumor nodes and a simultaneous decrease in their content in the spleen.

## 3. Discussion

Recently, it has been well documented that cfDNA exists in blood in both healthy and diseased individuals, but the composition and concentration of cfDNA significantly differ between patients with cancer and healthy donors [31,32]. Some data on the characteristics of circulating cfDNA have also been accumulated from laboratory animals, mostly mice, both healthy and under various pathological conditions [33,34,35]. It is worth noting that cfDNA, being a component of the tumor microenvironment, can affect both tumor cells themselves and surrounding cells; however, the potentiating effect of cfDNA on tumor progression remains unexplored.

Here, we investigated the effect of cfDNA generated by B16 melanoma cells and L929 fibroblasts on the properties of B16 cells in vitro and in vivo. Recently, we found that the progression of B16 melanoma is accompanied by an elevated concentration of cfDNA enriched with *L1* and *B1* tandem repeats and oncogene fragments in the blood of B16-transplanted mice [23]; therefore, we analyzed these in both genomic and cfDNA. It was shown that cfDNA composition reflected the composition of genomic DNA for both B16 and L929 cells, with some redistribution of fragment representation. Analysis of cfDNA_B16_ and cfDNA_L929_ compositions revealed that the abundance of *B1* and *L1* tandem repeats and oncogene fragments (*Myc* and *Ras*) was similar, which agrees well with the data of other researchers [36].

Unexpected results were obtained for *Myc*. Although gDNA_L929_ was characterized by low levels of *Myc* fragments, cfDNA_L929_ contained moderate levels of *Myc_ex1* and *Myc_ex3* and high levels of *Myc_ex2*, which exceeded those of cfDNA_B16_. An increase in the abundance of *Myc* copies and an elevated level of its expression is typical for various tumor cells [37]. Thus, these results raise the question of the origin of tumor-specific DNA in the bloodstream of tumor-bearing organisms. Moreover, our data suggest that the source of elevated *Myc* levels may be fibroblasts, particularly cancer-associated fibroblasts located in the tumor microenvironment.

Analyzing the presence of mitochondrial DNA in the cfDNA composition revealed mitochondrial DNA sequences only in cfDNA_B16_, whereas cfDNA_L929_ did not contain them. Mitochondrial DNA differs from genomic DNA in the presence of unmethylated CpG sequences. Moreover, there is an increased likelihood of the presence of oxidized and damaged DNA in mitochondrial DNA fragments due to the action of mitochondrial ROS. However, we did not detect 8-oxoG, a marker of damaged DNA as a result of oxidative stress, neither in the cfDNA_B16_ samples nor in the B16 conditioned medium; thus, further research is required.

The study of the effect of cfDNA_B16_ on B16 cell proliferation, apoptosis, and migration in vitro showed a decrease in the invasive potential of B16 cells, which appeared as a significant decrease in cell viability and migration rate as well as apoptosis initiation. An unexpected result was that cfDNA_L929_ increased the invasive potential of B16 cells by slightly increasing cell viability and significantly enhancing cell migration. The enhancement of malignant properties of B16 cells after treatment with fibroblast-derived cfDNA (cfDNA_L929_), which is expressed in the increase in cell migration in vitro and lung metastasis number in vivo, again raises the question of whether CAF-derived cfDNA contributes to metastasis.

In order to find ways in which cfDNA acts at the molecular level, we analyzed changes in the expression (mRNA level) of several DNases, oncogenes, and genes involved in key cellular processes after cfDNA treatment. We found that cells treated with both cfDNAs increased the expression of genes encoding secretory DNases, mitochondrial DNase, apoptotic DNase, and the oncogenes *Braf*, *Kras*, *Myc,* and *Trp53.* Interestingly, cfDNA_L929_ exhibited a stronger influence on *Myc* expression than cfDNA_B16_.

The observed increase in *Atp5If1* caused by cfDNA_L929_ suggests that the cells experienced oxidative stress because *Atp5If1* is involved in the modulation of mitochondrial pH and redox potential. In addition, a significant increase in the expression of *Endog*, whose product is involved in mtDNA cleavage in response to oxidative stress, after treatment with cfDNA_L929_, along with a decrease in the expression of *Cycs*, which serves as a key component of the mitochondrial electron transport chain, supports the idea of oxidative stress. Notably, despite the presence of cfDNA_L929_, there were no signs of apoptosis in B16 cells; moreover, our results indicated activation of cell viability under the action of this cfDNA (Figure 1B).

*Cdk4* and *Stat3* activated by both cfDNAs are implicated in the process of cellular division, which aligns with the findings regarding the proliferation activity of cfDNA_L929_. Interestingly, cfDNA_B16_ essentially decreased the expression of *Hmbg1*, which is involved in the organization of DNA, cell differentiation, and tumor cell migration. Both cfDNAs decreased *Rhoa* expression, overexpression of which is associated with tumor cell proliferation and metastasis [38]. These data correlated well with the observed decrease in the proliferation and migration of B16 cells under the action of cfDNA_B16_.

Thus, the mechanism of action of cfDNA on cells may consist of several stages. Changes in the ratio of nuclear DNA to mitochondrial DNA in the composition of cfDNA can contribute to the modulation of the effect of DNA on cells [39]. It has been shown that cfDNAs that differ in the ratio of mtDNA to genomic DNA differentially activate the TLR9/TLR4 pathways, followed by signal transduction to the NF-κB pathway [40,41]. The presence of mtDNA in cfDNA_B16_ may alter TLR9/TLR4 pathway signaling and lead to the effects we observed. Activation of the TLR9 pathway in some cases can lead to the secretion of TNFa, as, for example, in the case of lung adenocarcinoma A549 cells [42]. According to some data, externally administered TNFa can lead to apoptosis of tumor cells [43,44], which correlates with the data obtained on the apoptotic action of cfDNA_B16_.

In vivo, we also observed stimulation of tumor progression after preincubation of B16 cells with cfDNA_L929_ and inhibition of tumor growth after preincubation with cfDNA_B16_. The activation of tumor progression was visible both at the primary level by the size of the tumor node and the number of metastases and at the secondary level by the rate of tumor growth, the number of dividing cells, and the weight loss of laboratory animals. This could also be seen through indirect signs of disease severity, such as the concentration of cfDNA in the blood, blood DNase activity, the number of neutrophils in the blood (with a subcutaneous injection of tumor cells), and the number of neutrophils in the spleen.

Tumor and metastasis progression is accompanied by a pathological increase in the concentration of cfDNA in the blood of patients, which, in turn, is often accompanied by a decrease in DNase activity in the blood serum [45,46,47]. In general, according to the totality of data in the literature and the results of our studies, the concentration of cfDNA and DNase activity in the blood can serve as indicators of tumor progression with a high correlation coefficient (Figure 4).

We analyzed the relationships between the primary tumor size or metastases and other parameters, such as weight loss, cfDNA concentration, blood DNase activity, number of immune cells in the blood, tumor tissues, and spleen, and the amount of mitosis in the metastatic foci in the lungs, using regression analysis (Figure 4). The strongest negative correlation was found between blood DNase activity and the number of lung metastases and tumor size in both tumor models (R^2^ = 0.8–0.9). Another strong correlation was found between blood cfDNA concentration and metastasis number (R^2^ = 0.88), which correlates well with previous data [23]. It is important to note that blood DNase activity appears to be the most promising parameter for assessing, first, the severity of the disease and, second, treatment efficacy.

We found, as expected, a good correlation between weight loss and metastasis number (R^2^ = 0.7), although a poor correlation was observed between the same parameter and tumor size (R^2^ = 0.2) (Figure 4). We also observed a moderate correlation (R^2^ = 0.54 and 0.62) between the number of mitoses and both main tumor parameters (size and metastasis number). However, we could not find even a moderate correlation between the number of immune cells in the blood, spleen, and tumor tissue and the main tumor parameters. Nevertheless, we did find a poor but interesting negative correlation between tumor size and total number of tumor-infiltrated lymphocytes (R^2^ = −0.36), tumor-infiltrated neutrophils (R^2^ = −0.19), and tumor-infiltrated macrophages (R^2^ = −0.22). In addition, a negative correlation was observed between the number of lung metastases and spleen neutrophil count (R^2^ = −0.38).

According to various studies, tumor development is often accompanied by an increase in granulocytes in the blood and tumor tissues [48]. During tumor development caused by B16 cells treated with cfDNA_L929_, we observed an increase in the number of neutrophils in the blood, a decrease in their number in the spleen, and an increase in neutrophil infiltration of the tumor. These observations contrasted with tumor development by B16 cells treated with cfDNA_B16_; we observed inhibition of primary tumor growth and, accordingly, a decrease in neutrophil number in the blood and tumor and their accumulation in the spleen.

In summary, we found unexpected results: fibroblast-derived cfDNA stimulates the proliferation and migration of B16 melanoma cells, whereas autologous cfDNA_B16_ has a negative impact. The data we obtained indirectly indicate the importance of fibroblasts in the tumor microenvironment for tumorigenesis in general. Obviously, further research into this observation is required to reveal the mechanisms of various effects of cfDNA not only of tumor origin but also from the tumor microenvironment. Moreover, studying the composition and structural features of cfDNA of different origins and searching for underlying response mechanisms within tumor cells themselves is required.

## 4. Materials and Methods

### 4.1. Cell Cultures and Tumor Strains

Mouse fibroblast L929 (NCTC clone 929) and mouse B16-F10 melanoma cell lines were acquired from the Institute of Cytology of the Russian Academy of Sciences (St. Petersburg, Russia). The two cell lines were cultured in DMEM (Dulbecco’s Modified Eagle Medium; Thermo Fisher Scientific, Waltham, MA, USA) with 10% fetal bovine serum (FBS; HyClone, Washington, WA, USA) and 1% antibiotic–antimycotic solution (10 mg/mL streptomycin, 10,000 U/mL penicillin, and 25 μg/mL amphotericin (MP Biomedicals, Santa Ana, CA, USA)) at 37 °C in a humidified atmosphere containing 5% CO_2_ (standard conditions) and regular passages to maintain exponential growth.

### 4.2. Mice

Male, 10–14-week-old C57Bl/6 (hereinafter, C57Bl) mice were purchased from the vivarium of ICBFM SB RAS (Novosibirsk, Russia). Mice were kept in plastic cages (ten animals per cage) under standard daylight conditions. Water and food were provided ad libitum. All animal procedures were performed in accordance with the recommendations for the proper use and care of laboratory animals (ECC Directive 2010/63/EU) [49]. The experimental protocols were approved by the Committee on the Ethics of Animal Experiments of the Administration of the Siberian Branch of the Russian Academy of Sciences (Novosibirsk, Russia) (ethical approval number: 49; 23 May 2019), and all efforts were made to minimize suffering.

At the start of the experiments, the animal weight (mean ± SD) was 20.2 ± 1.5 g.

### 4.3. Obtaining a Conditioned Medium

L929 and B16 cells were cultured in 25 cm^2^ cell culture flasks until they reached approximately 90% confluence (up to 48 h). Once grown, the media were removed, and the cells were rinsed twice with PBS. Subsequently, fresh FBS-free media containing antibiotic–antimycotic solution was added. The cells were then incubated in standard incubation conditions for 24 h to allow the formation of the conditioned media. The conditioned media were then collected and centrifuged at 350× *g* for 10 min. The resulting supernatants were transferred to 2 mL tubes (each containing 2 mL of supernatant) and concentrated to volumes of 100 µL by evaporation. These concentrated supernatants were stored at −20 °C for cfDNA isolation.

### 4.4. Isolation of cfDNA and Genomic DNA

#### 4.4.1. Conditioned Medium

For the processing of the conditioned medium, 2 µL of RNase A per 1000 µL of evaporated supernatant was added to the samples (RNase A, DNase- and protease-free (10 mg/mL) (Sigma-Aldrich, Darmstadt, Germany)). The mixtures were then incubated at 65 °C for 60 min, followed by subsequent incubation at 37 °C for 15 min. cfDNA extraction was carried out by phenol (phenol-Tris-HCl, pH 8.0) and chloroform extraction methods, followed by concentration using the QIAquick Gel Extraction Kit (Qiagen, Germantown, MD, USA), following the manufacturer’s instructions. DNA was separated by centrifugation, and the resulting precipitate was washed with 80% ethanol, dried, dissolved in water, and stored at −20 °C.

The concentration of cfDNA was determined using a Qubit fluorometer (Invitrogen, Carlsbad, CA, USA) with a Quant-iT dsDNA HS Assay Kit (Thermo Fisher Scientific, Waltham, MA, USA), following the manufacturer’s instructions. The quality of cfDNA was assessed using a NanoDrop™ ND-1000 spectrophotometer (ThermoFisher Scientific, Waltham, MA, USA) and verified using a 1% agarose gel.

#### 4.4.2. Blood Serum

cfDNA was extracted from blood serum using the DNeasy Blood & Tissue Kit (Qiagen, Germantown, MD, USA) according to the manufacturer’s instructions. The concentration of cfDNA was quantified using a Qubit fluorometer, as described previously.

#### 4.4.3. Cells

L929 and B16 cells were cultured in 25 cm^2^ cell culture flasks until they reached approximately 90% confluence (up to 48 h). The medium was then removed, and the cells were detached using Try-pLE™ (Thermo Fisher Scientific, Waltham, MA, USA), following the manufacturer’s guidelines. After rinsing with PBS, the cells were collected by centrifugation at 300× *g* for 5 min. To the cell pellet, 50 µL of 0.25% trypsin in PBS (MP Biomedicals, Santa Ana, CA, USA) was added and incubated for 2 min. Subsequently, 500 µL of PBS supplemented with 10% FBS was added, and the cells were collected by centrifugation at 350× *g* for 10 min, following which the supernatant was discarded. The cells were lysed in a solution containing 100 mM Tris-HCl (pH 8.0), 5 mM EDTA, 200 mM NaCl, 0.2% SDS, and 10 ng/mL proteinase K (Thermo Fisher Scientific, Waltham, MA, USA) at 65 °C for 4 h with continuous mixing. Genomic DNA was then isolated using a phenol–chloroform method followed by ethanol precipitation. The concentration of DNA was determined using a Qubit™ fluorometer, and the quality of DNA was assessed using a NanoDrop™ ND-1000 spectrophotometer and verified using a 1% agarose gel.

### 4.5. Isolation of Total RNA

The cells were detached using TrypLE™ according to the manufacturer’s instructions, washed with PBS, and collected by centrifugation at 300× *g* for 5 min. RNA isolation from the cells was carried out using Rizol (diaGene, Moscow, Russia) according to the manufacturer’s guidelines. The quality and concentration of the isolated RNA were evaluated using a NanoDrop™ D-1000 spectrophotometer.

### 4.6. Serum Preparation

Blood serum was prepared from whole blood collected in standard test tubes through clot formation at 37 °C for 30 min, followed by overnight incubation at 4 °C, clot removal, and centrifugation at 4000 rpm at 4 °C for 20 min to eliminate cell debris. Serum samples were stored at −70 °C.

### 4.7. Blood Analysis

Blood samples were collected via retro-orbital sinus puncture using heparinized microcapillary tubes. Hematological parameters, such as total and differential leukocyte counts, were assessed using a hematology analyzer (MicroCC20Vet; High Technology Inc., North Attleborough, MA, USA).

### 4.8. Assessment of DNase Activity in Blood Serum

The total DNase activity of blood serum was measured in a cleavage reaction of plasmid pHIV kindly provided by Dr. Y. Staroseletz (this institute) [50]. The reaction mixture of 30 μL, consisting of 1 μL of serum and 0.5 μg of pHIV, was incubated at 30 °C for 5–30 min. The reaction was quenched by adding EDTA to reach a final concentration of 2.5 mM. Subsequently, the mixtures were incubated at 65 °C for 10 min, followed by extraction with phenol (pH 8.0) and chloroform. The resulting cleavage products were analyzed by electrophoresis on a 1% agarose gel stained with ethidium bromide. The effective cleavage rate constants (k_eff_) were determined using the equation: Pt = P∞ × (1 − exp^−(keff×t)^), where Pt and P∞ represent the fraction of the substrate cleaved at time t and at the end point, respectively [23].

### 4.9. Cell Viability Assay

Cell viability was assessed using the MTT test. B16 cells were seeded in a 96-well plate at a density of 8 × 10^3^ cells per well in FBS-free DMEM with antibiotic–antimycotic solution, incubated under standard conditions for 12 h until 90% confluency, then treated with cfDNA_B16_ or cfDNA_L929_ (100 or 1000 ng/mL) and incubated for 24 h [23]. Subsequently, MTT solution (Thermo Fisher Scientific, Waltham, MA, USA) was added to the cells at a concentration of 0.5 mg/mL, and the cells were further incubated for 3 h and analyzed as described in [51]. The viability of the cells was calculated as a percentage relative to untreated control cells using the equation: (N_exp_/N_c_) × 100%.

### 4.10. Scratch Assay

B16 cells were seeded in a 6-well plate at a density of 1.5 × 10^6^ cells per well in FBS-free DMEM with antibiotic–antimycotic solution and incubated under standard conditions for 12 h until 90% confluency was reached. A 0.5 mm wide scratch was made in the cell monolayer, followed by washing with PBS and the addition of FBS-free DMEM supplemented with the antibiotics and cfDNAB16 or cfDNAL929 at concentrations of 100 or 1000 ng/mL. The cells were then incubated for 24 h under standard conditions. The migration of cells into the scratched area was monitored for 24 h using a Zeiss Primo Vert microscope (Zeiss, Oberkochen, Germany) and analyzed using ImageJ software version 12 [23].

### 4.11. Flow Cytometry

For apoptosis measurement, Annexin V-FITC/PI staining was used. In brief, B16 cells were seeded in a 24-well plate in FBS-free DMEM supplemented with antibiotic–antimycotic solution at a density of 0.5 × 10^6^ cells per well. After 12 h, the medium was replaced with fresh medium of the same composition, cfDNA_B16_ or cfDNA_L929_ were added at concentrations of 100 or 1000 ng/mL, and cells were incubated under standard conditions for 24 h. Cells were detached with TrypLE™ Express Enzyme with no phenol red (Thermo Fisher Scientific, Waltham, MA, USA). Apoptosis was analyzed using an Annexin V-FITC Apoptosis Staining/Detection Kit (Abcam, Cambridge, UK) according to the manufacturer’s protocol using a NovoCyte flow cytometer (ACEA Biosciences, San Diego, CA, USA).

TLR9/PI staining was used for TLR9 visualization. B16 cells were seeded into Petri dishes (d = 40) in FBS-free DMEM supplemented with antibiotic and antimycotic solution at a density of 3 × 10^6^ cells per dish and incubated under standard conditions. Cells were detached as described above and then divided into two parts: one part of the cells was immediately stained, and the other was stained after treatment with 0.5% Tween20 (Wuhan Servicebio Technology Co., Ltd., Wuhan, China) in PBS. Cells were stained with 3 μL of anti-mouse anti-TLR9-APC (1395510; SONI Biotechnology, San Jose, CA, USA) and 3 μL PI in PBS, with preliminary incubation in PBS with 1% FBS. The staining intensity was analyzed as described above.

### 4.12. qPCR Analysis of the Abundance of MGEs and Oncogene Fragments and Mitochondrial Genes in cfDNA

The abundance of MGE and oncogene sequences in DNA samples was determined by qPCR. Amplification was performed in 20 μL of reaction mixture containing 0.1–0.5 ng of DNA, SYBR-Green-containing Bio Master CorHS-qPCR (BiolabMix, Novosibirsk, Russia), and 0.6 μM of each forward and reverse specific primers for MGE (*B1_mus2* and *L1td1*), oncogenes (*Myc* and *Ras*), and mitochondrial genes (*mt-Nd3* and *mt-Co3*). The primer sequences are listed in Appendix A. PCR was performed using a CFX96 Touch Real-Time PCR Detection System (Bio-Rad Laboratories Inc., Hercules, CA, USA). The reaction conditions were as follows: 95 °C, 6 min; 95 °C, 15 s; 60 °C, 20 s; 70 °C, 60 s; 50 cycles. *Gapdh* was used as the reference gene.

### 4.13. Analysis of Gene Expression Using RT-qPCR

cDNA was prepared in 40 μL of reaction mixture containing 2 μg total cellular RNA, 5× RT buffer mix (Biolabmix), 100 U reverse transcriptase MMuLV-RH (Biolabmix, Novosibirsk, Russia), and 0.05 μM primer dT15. Reverse transcription was carried out as follows: 42 °C, 1 h; 70 °C, 10 min.

The reaction mixture for qPCR (25 μL) contained 0.1–0.5 ng of cDNA, BioMaster HS-qPCR SYBR (BioabMix), and 0.6 μM of each of the forward and reverse specific primers and probes (Appendix A). The reaction conditions were as follows: 95 °C, 6 min; 95 °C, 15 s; 60 °C, 20 s; 70 °C, 60 s; 45 cycles. *Actb* was used as a reference.

### 4.14. Tumor Implantation and Design of Animal Experiments

B16 cells were grown in 25 cm^2^ cell culture flasks in IMDM medium supplemented with 10% FBS and antibiotic–antimycotic solution until ~90% confluence was reached (up to 48 h). The medium was discarded, and the cells were washed twice with PBS. cfDNA_B16_ or cfDNA_L929_ in FBS-free IMDM medium supplemented with antibiotic–antimycotic solution was added to the cells at a final concentration of 100 ng/mL, and the cells were incubated under standard conditions for 24 h. B16 cells incubated in FBS- and antibiotic-free IMDM were used as controls. The cells were harvested, washed twice with PBS, and resuspended in saline buffer. To generate a model of melanoma B16 with primary nodes, 0.1 mL (10^6^ cells/mL) was injected s.c. into the withers of mice. To generate metastatic melanoma B16, 0.2 mL (5 × 10^5^ cells/mL) was injected into the lateral tail veins of mice.

Six groups of mice were formed: 1, 2, and 3—intact B16 cells (control), B16 pretreated with cfDNA_L929_, and B16 pretreated with cfDNA_B16_, respectively, implanted s.c.; 4, 5, and 6—intact B16 cells (control), B16 pretreated with cfDNA_L929_, and B16 pretreated with cfDNA_B16_, respectively, implanted i.v. In both experiments, the mice were kept for 22 days, and weight measurements were taken every 3–4 days. In an experiment with a primary tumor, tumor size was determined on alternate days with caliper measurements in three perpendicular dimensions. Tumor volumes were calculated as V = (π/6 × length × width × height).

In both experiments, on day 22, blood samples were collected from the retro-orbital sinus into test tubes with heparin and into ordinary test tubes for the preparation of blood serum. Spleens were harvested and neutrophils were isolated using positive selection as described by Sounbuli et al. [52]. The mice were then sacrificed, and the lungs occupied by metastases were isolated and fixed in 4% neutral-buffered formaldehyde (BioVitrum, St. Petersburg, Russia) for subsequent histological analysis. Surface metastases were counted using a binocular microscope.

### 4.15. Histology and Morphometry

Primary tumor nodes and lungs with metastases were collected and fixed in 10% neutral-buffered formalin (BioVitrum, Moscow, Russia), dehydrated in ascending ethanols, and embedded in HISTOMIX paraffin (BioVitrum, Moscow, Russia). Paraffin sections (up to 5 μm) were sliced on a Microm HM 355S microtome (Thermo Fisher Scientific, Waltham, MA, USA) and stained with hematoxylin and eosin. Images were examined and scanned using an Axiostar Plus microscope equipped with an Axiocam MRc5 digital camera (Zeiss, Oberkochen, Germany) at magnifications of ×400 and ×1000 (oil).

Morphometric analysis of tumor tissue and lung metastases included evaluation of the numerical density (Nv) of mitoses, indicating the number of mitotic events in the square unit—3.2 × 10^6^ μm^2^ in this case. Differential cell counting in tumor tissue included calculation of the number of cells (macrophages, lymphocytes, and neutrophils) forming reactive infiltration of primary tumor nodes according to the formula: differential number of cells/total number of cells × 100%. Five random fields were studied in each specimen, forming 50 random fields for each group of mice in total.

### 4.16. Statistical Analysis

Correlation analysis was carried out using the multiple regression order correlation coefficient (R^2^), which indicates the correlation between observed and predicted values of the outcome variable. The strength of the relationship between parameters (variables) was interpreted based on the regression analysis coefficient values: 0.01 ≤ r or R^2^ < 0.3 indicated a weak correlation, 0.3 ≤ r or R^2^ < 0.7 indicated a moderate correlation, and 0.70 ≤ r or R^2^ ≤ 0.99 indicated a strong correlation. Positive and negative values denoted positive and negative correlations.

All experiments were replicated three times. Data from the MTT and scratch assays were statistically analyzed using a two-tailed unpaired Student’s *t*-test. Parameters such as MII, metastasis number, DNase activity, cfDNA concentration, and PCR data were statistically evaluated using one-way ANOVA. Post hoc analysis was performed using a Tukey test, with statistical significance set at *p* < 0.05. The statistical software STATISTICA version 10.0 was employed for data analysis.

## Figures and Tables

**Figure 1 ijms-25-05304-f001:**
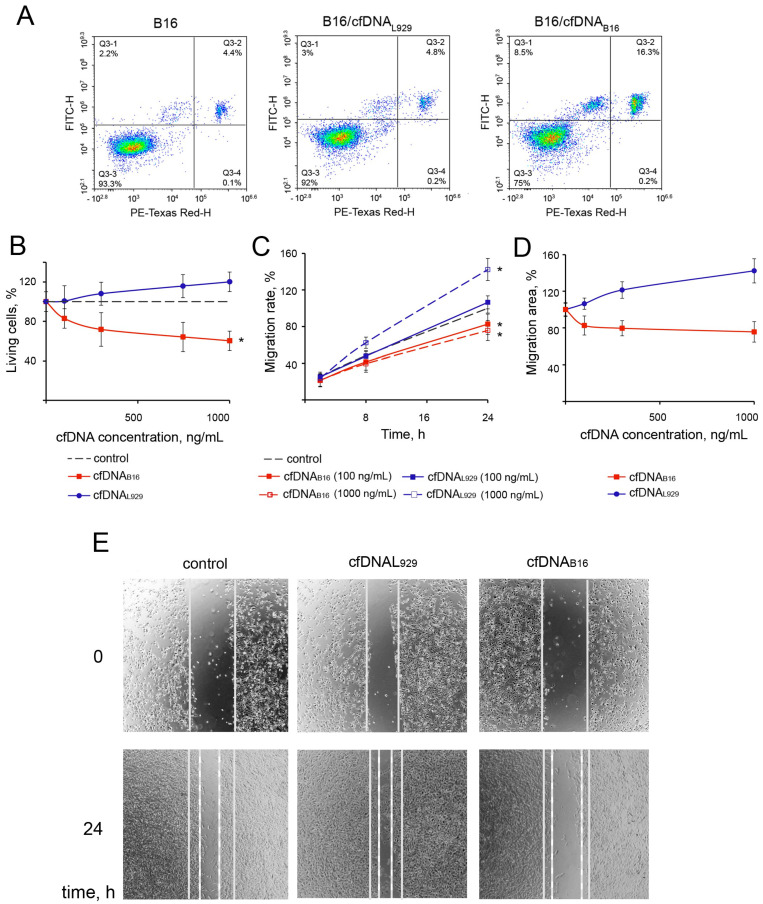
Effect of cfDNA_B16_ and cfDNA_L929_ on apoptosis, viability, and migration of B16 cells. (**A**) Flow cytometry data. (**B**) Viability of B16 cells. (**C**,**D**) Migration rate and migration area of B16 cells. Scratch assay. The data of the MTT test and scratch assay are presented as means ± SDs. (**E**) Scratch healing by B16 cells after incubation with cfDNA (representation photographs, 4× magnification). All experiments were reproduced in six replicates. Data were statistically analyzed using a one-way ANOVA with a post hoc Tukey test. * *p* < 0.05—statistical difference between control and experimental groups. Solid lines—scratch boundaries at time 0; dotted lines—the cell front boundaries at 24 h. Control—untreated B16 cells. B16 cells were incubated with cfDNA_B16_ and cfDNA_L929_ at concentrations of 100 and 1000 ng/mL. For the experiments in (**A**,**E**), a concentration of cfDNA 1000 ng/mL was used.

**Figure 2 ijms-25-05304-f002:**
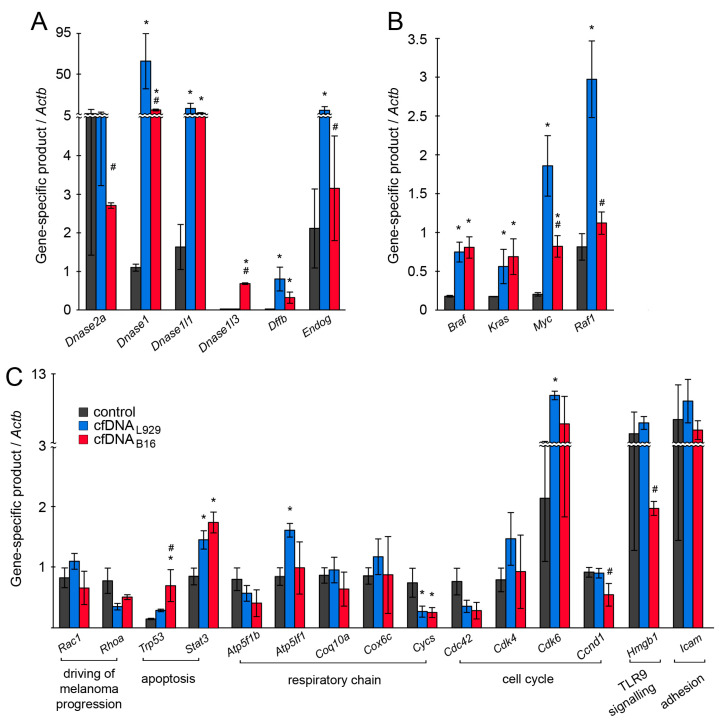
mRNA levels in B16 cells exposed to cfDNA. The data of RT-qPCR. The levels of mRNA encoded DNases (**A**), oncogenes (**B**), and genes involved in key cellular processes (**C**). The reference gene was *Actb*. All experiments were reproduced in eight replicates. Control: intact cells. The data were analyzed using a Student’s *t*-test and are presented as means ± SDs. * *p* < 0.05—statistical difference between control and experimental groups; # *p* < 0.05—statistical difference between experimental groups.

**Figure 3 ijms-25-05304-f003:**
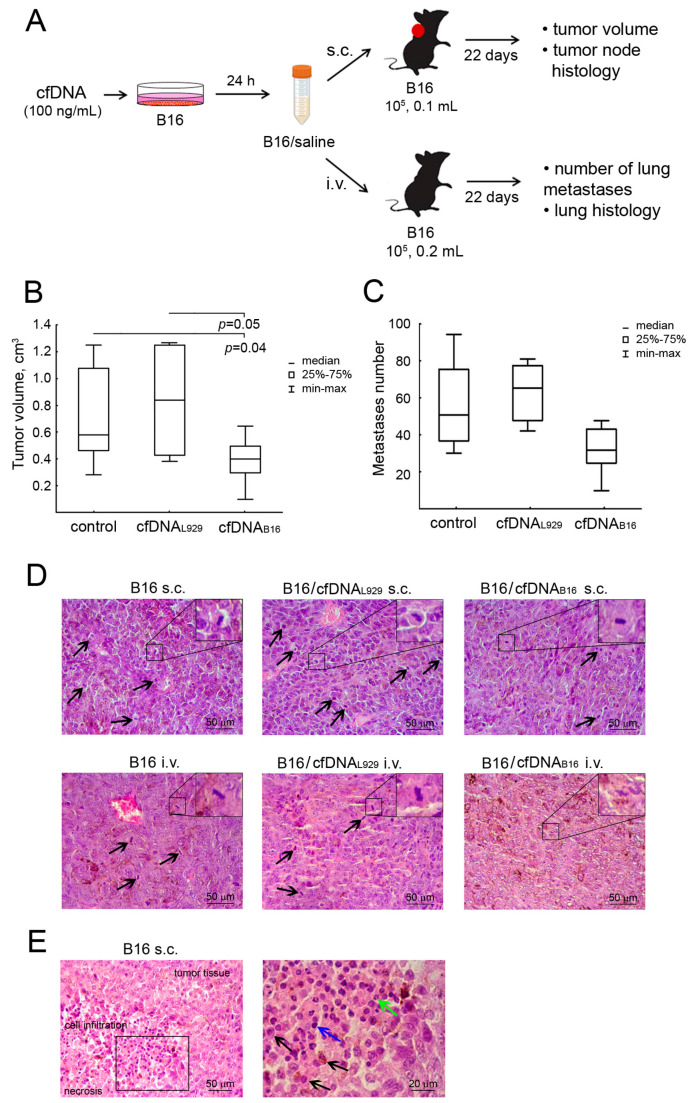
The effect of cfDNA on B16 melanoma node growth and metastasis development. (**A**) Scheme of an experiment. B16 cells were pretreated with cfDNA_L929_ and cfDNA_B16_ (100 ng/mL) for 24 h, harvested, resuspended in saline buffer, and implanted subcutaneously (s.c., 10^5^ cells, 0.1 mL) or intravenously (i.v., 10^5^ cells, 0.2 mL) into C57Bl mice. All experiments were reproduced in triplicate. (**B**) Tumor volume. (**C**) Number of surface lung metastases. The data are presented as medians. Data were statistically analyzed using a one-way ANOVA with a post hoc Tukey test. Statistical significance is *p* < 0.05. (**D**) Representative histological images of B16 melanoma primary tumor nodes (upper panel) and lung metastases (bottom panel). Mitoses are indicated by black arrows. Typical examples of individual mitotic events are shown in the upper right corner. Hematoxylin and eosin staining. Original magnification: ×400. (**E**) Histological structure of the B16 melanoma primary tumor node (left panel) and cell infiltration of tumor tissue (right panel). The black box shows areas that were examined further at higher magnification. Black arrows indicate macrophages, blue arrows indicate lymphocytes, and green arrows indicate neutrophils. Hematoxylin and eosin staining. Original magnification: ×400 and ×1000.

**Figure 4 ijms-25-05304-f004:**
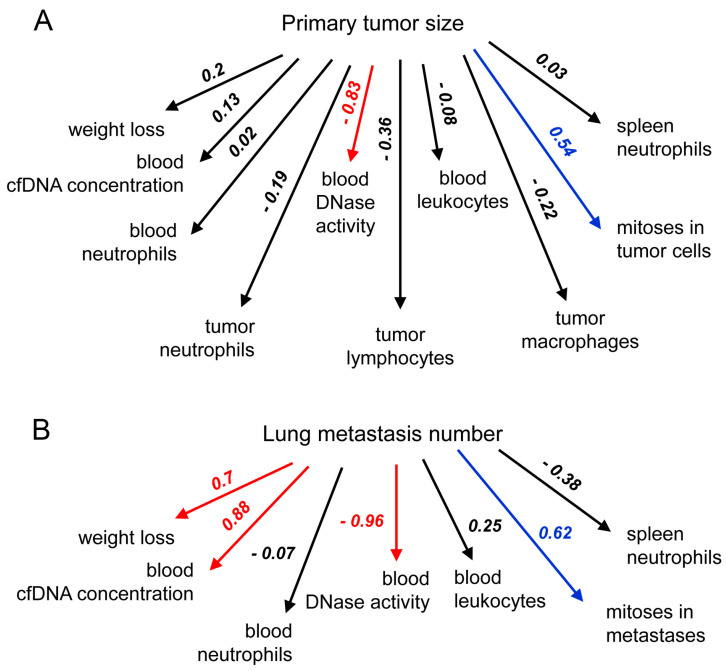
Correlations between lung metastasis number or primary tumor size and other potential tumor characteristics. Multiple regression order correlation coefficients (R^2^) reflecting the statistical relationships between the studied parameters were evaluated for mice with subcutaneously (**A**) and intravenously (**B**) implanted B16. Red arrows indicate strong positive or negative correlations (0.70 ≤ R^2^), blue arrows indicate moderate correlations (0.3 ≤ R^2^), and black arrows indicate weak correlations (0.01 ≤ R^2^).

**Table 1 ijms-25-05304-t001:** Abundance of oncogene and MGE fragments in genomic DNA and cfDNA from conditioned media of B16 and L929 cells.

Gene	gDNA_B16_	cfDNA_B16_	gDNA_L929_	cfDNA_L929_
*B1_mus1*	8.0 ± 0.2	7.5 ± 0.2	3.3 ± 0.2	4.2 ± 0.2
*B1_mus2*	7.9 ± 0.1	9.7 ± 0.1	2.8 ± 0.3	5.0 ± 0.4
*L1_mus1*	2.6 ± 0.1	2.4 ± 0.1	3.33 ± 0.9	2.9 ± 0.4
*L1td1_orf1*	1.5 ± 0.3	1.3 ± 0.01	0.01 ± 0.001	2.2 ± 0.1
*L1td1_orf2*	10.6 ± 1.1	3.3 ± 0.5	8.5 ± 0.9	3.1 ± 0.2
*Myc_ex1*	8.0 ± 0.2	12 ± 1.1	1.7 ± 0.2	15.5 ± 1.0
*Myc_ex2*	13.0 ± 0.1	13.2 ± 2.0	3.3 ± 0.2	93.0 ± 6.0
*Myc_ex3*	8.0 ± 0.3	9.5 ± 1.5	1.7 ± 0.2	16.0 ± 3.1
*Ras*	1.0 ± 0.1	3.2 ± 0.2	1.0 ± 0.1	0.9 ± 0.3
*mt-Nd3*	n.d.	7.8 ± 1.5	0.01 ± 0.001	1.2 ± 0.3
*mt-Co3*	n.d.	n.d.	n.d.	n.d.

The abundance of oncogene and MGE fragments was determined by qPCR. Data are presented as means ± SDs. n.d.—not detected. The number of gene copies was normalized to the number of *Gapdh* copies.

**Table 2 ijms-25-05304-t002:** Effect of cfDNA on the characteristics of mice with subcutaneously implanted melanoma B16.

Parameters	Healthy	Group 1	Group 2	Group 3
Tumor volume, cm^3^	-	0.6 ± 0.2	0.8 ± 0.2	0.4 ± 0.1 *#
Weight loss, %	-	2.7 ± 1.3	3.9 ± 2.1	0.1 ± 0.05 #
cfDNA concentration in blood serum, ng/mL	567 ± 201	1267 ± 146 *^α^	1242 ± 384	796 ± 230
DNase activity of blood serum, ×10^−3^ s^−1^	0.6 ± 0.1	0.4 ± 0.2	0.2 ± 0.2	0.6 ± 0.1 #
Blood leukocytes, ×10^8^/mL	12.6 ± 2.1	15.2 ± 3.7	11.7 ± 2.5	9.8 ± 4.5
Blood neutrophils, %	7.8 ± 1.8	12.0 ± 4.4	27.3 ± 7.5	3.2 ± 2.9
Mitoses in tumor tissue, Nv	-	3.8 ± 0.9	6.9 ± 2.4 *	3.0 ± 1.5 #
Neutrophils in tumor tissue, %	-	7.9 ± 5.2	9.8 ± 7.2	6.0 ± 2.5
Lymphocytes in tumor tissue, %	-	9.2 ± 2.4	10.1 ± 2.8	12.2 ± 2.9
Macrophages in tumor tissue, %	-	85.1 ± 3.5	79.0 ± 5.7	83.0 ± 4.5
Neutrophils in spleen, ×10^6^	4.5 ± 3.2	3.7 ± 1.5	3.9 ± 1.4	4.0 ± 1.0

Group 1—intact B16 cells (control); group 2—B16 cells pretreated with cfDNA_L929_; group 3—B16 cells pretreated with cfDNA_B16_. B16 cells were implanted s.c. Data are presented as means ± SDs. ^α^—statistical difference between healthy and control mice (group 1); *—statistical difference between control (group 1) and groups 1 and 2; #—statistical difference between groups 2 and 3. Nv—numerical density.

**Table 3 ijms-25-05304-t003:** Effect of cfDNA on the characteristics of mice with intravenously implanted melanoma B16.

Parameters	Healthy	Group 4	Group 5	Group 6
Metastasis number	-	56 ± 25	69 ± 14	39 ± 13
Weight loss, %	-	7.0 ± 7.0	19.5 ± 4.0	4.3 ± 4.2 #
cfDNA concentration in blood serum, ng/mL	567 ± 152	1122 ± 257 ^α^	1395 ± 239	760 ± 133 *#
DNase activity of blood serum, ×10^−3^ s^−1^	0.6 ± 0.1	0.4 ± 0.1 ^α^	0.2 ± 0.1 *	0.6 ± 0.1 *#
Blood leukocytes, ×10^8^/mL	7.6 ± 2.2	9.5 ± 2.9	11.6 ± 4.5	9.0 ± 3.8
Blood neutrophils, %	7.8 ± 2.1	40.0 ± 22.1	37.8 ± 17.6	35.2 ± 20.9
Mitoses in tumor tissue, Nv	-	2.9 ± 1.9	3.4 ± 1.0	0.5 ± 0.5 *#
Neutrophils in spleen, ×10^6^	4.5 ± 1 2	4.5 ± 2.4	3.7 ± 2.4	4.5 ± 2.3

Group 4—intact B16 cells (control); group 5—B16 cells pretreated with cfDNA_L929_; group 6—B16 cells pretreated with cfDNA_B16_. B16 cells were implanted i.v. Data are presented as means ± SDs. ^α^—statistical difference between healthy and control mice (group 1); *—statistical difference between control (group 1) and groups 1 and 2; #—statistical difference between groups 2 and 3. Nv—numerical density.

## Data Availability

The data are available from the corresponding author upon e-mail request.

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
