# Peer review of "Tumor- and Fibroblast-Derived Cell-Free DNAs Differently Affect the Progression of B16 Melanoma In Vitro and In Vivo"

_ijms, 2024, doi:10.3390/ijms25105304_

Round 1

Reviewer 1 Report

Comments and Suggestions for Authors

The manuscript by Filatova et al. provides compelling evidence that sheds light on the potential role of DNA fibroblasts in the tumor microenvironment, specifically in tumor progression as well as the formation of new tumor foci through the transformation of normal cells.

The manuscript is very nicely written. I have a few suggestions for polishing the manuscript.

Comments:

  1. The manuscript should be checked by a native English speaker.
  2. In Figure 1B, the authors should add a few more intermediate concentrations. Also, they should indicate IC50 values.
  3. All the data should be represented in the form of mean ± SD instead of mean ± SEM.
  4. The authors should discuss the results more precisely and in detail. The authors only stated the results, but they need to add the significance behind the results. 
  5. Figure 3 is a very important data of the manuscript, but it is not discussed precisely. The authors should explain their results in a more in-depth manner.
  6. The authors showed that cfDNAB16 decreased the invasive potential of B16 cells and cfDNAL929 increased it. But the authors did not explain the reasons behind it. They did test some signature markers, but they did not exactly pinpoint the reasons behind this phenomenon. The authors should study differential gene expression in the absence and presence of cfDNAL929 and cfDNAB16 to further dissect the molecular mechanism behind this.
  7. The authors should add a graphical abstract of their results for a broader audience.

Comments on the Quality of English Language
  1. The manuscript should be checked by a native English speaker.

Author Response

The manuscript by Filatova et al. provides compelling evidence that sheds light on the potential role of DNA fibroblasts in the tumor microenvironment, specifically in tumor progression as well as the formation of new tumor foci through the transformation of normal cells.

The manuscript is very nicely written. I have a few suggestions for polishing the manuscript.

Comments:

The manuscript should be checked by a native English speaker.

We have rewritten part of the manuscript in accordance with your comments and the comments of other reviewers. The corrected version was checked by a native English speaker and using the online service Trinka.

In Figure 1B, the authors should add a few more intermediate concentrations. Also, they should indicate IC50 values.

We added information about intermediate concentration. However, we did not present our data as IС50, as this may confuse the reader, because IC50 is usually used to describe the toxic effects of therapeutic agents, and we do not represent the effect of cfDNA as therapeutic.

All the data should be represented in the form of mean ± SD instead of mean ± SEM.

We corrected with data in Figure 1, Figure 2 and all Tables according to your recommendations (marked by red).

The authors should discuss the results more precisely and in detail. The authors only stated the results, but they need to add the significance behind the results. 

We performed several experiment to explain our data (RT-PCR to measure mitochondrial DNA in cfDNAs, 8-oxo-guanine ELISA, and TLR9 staining in B16 cells), and modified Results and Discussion (please see, marked by red).

Figure 3 is a very important data of the manuscript, but it is not discussed precisely. The authors should explain their results in a more in-depth manner.

We modified Results and Discussion, please see, marked by red.

The authors showed that cfDNAB16 decreased the invasive potential of B16 cells and cfDNAL929 increased it. But the authors did not explain the reasons behind it. They did test some signature markers, but they did not exactly pinpoint the reasons behind this phenomenon. The authors should study differential gene expression in the absence and presence of cfDNAL929 and cfDNAB16 to further dissect the molecular mechanism behind this.

We supplemented our study by measuring the contribution of mitochondrial DNA to the composition of cfDNA, measured the content of 8-oxo-guanine in cfDNAs and also investigated the presence of TLR9 receptors in B16 cells. Obtained data together with increase of the expression of Tp53 allow us to hypothesize that the expression of TNFa could be activated through TLR9 pathways under the action of cfDNAB16, which could promote apoptosis of B16 cells. We agree that differential gene expression in the absence and presence of L929 cfDNA as well as other cfDNAs should be conducted, but in our opinion such a study is beyond the scope of this article. We believe that this study should be carried out on the scale of a whole-exome or full-proteomic study and should be the material of another large article.

The authors should add a graphical abstract of their results for a broader audience.

As you recommended, we presented our observations and assumptions in the form of a graphic abstract.

Reviewer 2 Report

Comments and Suggestions for Authors

In this article, Filatova et al. showed that cell-free DNA (cfDNA) originated from different sources have diverse outcome on the growth of B16 murine melanoma cells. cfDNA from fibroblast have more adverse effect on B16 cells while cfDNA from tumors have negative impact on B16 cells growth and metastasis. The study is well designed, and the experiments are carefully performed. Some further clarifications are needed to strengthen the rational of the study and the conclusions.

Comments are as follows:

 Comments:

1.Why HMGA2 was chosen? What is the outcome means for this study?

2. In line 133, the conc. is stated 100ng/ml. But for Fig1A, the authors used 1000ng/ml. Can the authors clarify these discrepancies?       

3. For figure 2 what is the conc. of cfDNA was used? Can the authors show what is the effect when both the conc. is used?

4. For Figure 2 the authors performed statistics only between control vs. cfDNA treatment groups. The authors should also show statistics between the cfDNA_L929 vs. cfDNA_B16.

5. For figure 3 the authors limited their study to day 22. It is important to show whether different cfDNA can impact the survival of the mice? If cfDNA_B16 can improve the survival, then it will be worth following.

6. One of the major drawbacks of this manuscript is the authors pointed out the observations without providing any mechanisms. It will be helpful if the authors showed some form of mechanism. For example, CDK6 was highly upregulated in cfDNA_L929. If we add Palbociclib which inhibit both CDK4/6, whether it can improve the survival of the mice or at least increase apoptosis (https://pubmed.ncbi.nlm.nih.gov/34145036/). The authors need to bring a cohesiveness in the discussion section to improve the manuscript.

Author Response

Reviewer 2

In this article, Filatova et al. showed that cell-free DNA (cfDNA) originated from different sources have diverse outcome on the growth of B16 murine melanoma cells. cfDNA from fibroblast have more adverse effect on B16 cells while cfDNA from tumors have negative impact on B16 cells growth and metastasis. The study is well designed, and the experiments are carefully performed. Some further clarifications are needed to strengthen the rational of the study and the conclusions.

 Comments:

Why HMGA2 was chosen? What is the outcome means for this study?

Previously, we examined the cfDNA profile of the blood serum of mice with LLC using NGS sequencing. We found that the abundance of Hmga2, Fos, and Jun increased slightly with disease progression. We assumed that oncogene level in melanoma-derived cfDNA would also change during the development of the disease, but not as dramatically as, for example, SINEs. However, these sequences were literally not detected in any type of DNA examined. Initially, we wanted to leave these fragments as a “standard of absence”. However, from your recommendations we considered that this information may be superfluous for the article, and we have removed it.

In line 133, the conc. is stated 100 ng/ml. But for Fig1A, the authors used 1000ng/ml. Can the authors clarify these discrepancies?

The mistake has been corrected. Data are given for a concentration of 1000 ng/mL.

For figure 2 what is the conc. of cfDNA was used? Can the authors show what is the effect when both the conc. is used?

Figure 2 shows data for a concentration of 100 ng/mL. Since cells undergo apoptosis at 1000 ng/mL, we could not use this concentration due to high RNA degradation in the samples for RNA isolation (the resulting RNA was of significantly lower quality). And for the same reason this concentration was not apply for in vivo experiments.

For Figure 2 the authors performed statistics only between control vs. cfDNA treatment groups. The authors should also show statistics between the cfDNA_L929 vs. cfDNA_B16.

Figure 2 and figure’s legend were corrected.

For figure 3 the authors limited their study to day 22. It is important to show whether different cfDNA can impact the survival of the mice? If cfDNA_B16 can improve the survival, then it will be worth following.

We acted within the framework of the Bioethical Commission specified in the article. According to the commission's instructions, mice should be sacrificed when there is a significant decrease in quality of life, in particular when the tumor volume reaches 1 cm3. According to numerous studies, including one’s of our group, this volume in the control group is achieved by 21 days. To allow comparison of experiments with the B16 metastatic model and the B16 tumor model, all mice were withdrawn from the experiment on day 21.

One of the major drawbacks of this manuscript is the authors pointed out the observations without providing any mechanisms. It will be helpful if the authors showed some form of mechanism. For example, CDK6 was highly upregulated in cfDNA_L929. If we add Palbociclib which inhibit both CDK4/6, whether it can improve the survival of the mice or at least increase apoptosis (https://pubmed.ncbi.nlm.nih.gov/34145036/). The authors need to bring a cohesiveness in the discussion section to improve the manuscript

We have supplemented and expanded the Discussion section, focusing specifically on the possible mechanisms of the observed effects. We designed graphical abstract with possible mechanism of cfDNA action. We believe that such a study of the contribution of specific genes to the development of processes is absolutely necessary; it should become the material for future research. Moreover, we believe that experiments using selective inhibitors and antagonists should be preceded by a whole-exome or full-proteomic study.

Reviewer 3 Report

Comments and Suggestions for Authors

Please see attached for comments and suggestions.

Comments on the Quality of English Language

English writing is good in general.

Author Response

Reviewer 3

In this manuscript, the authors investigated the effects of tumor and normal cell originated cell-free DNAs on tumor cells by measuring the resulting cell proliferation, apoptosis and migration level and the expression level of certain genes. The authors also assessed the cfDNA treated tumor volume, metastases level, and other related characteristics in vivo. Below are my comments and questions.

Major concerns

The observation that normal cell originated cfDNA stimulates tumor while tumor cell originated cfDNA inhibits tumor is a major novel finding of this manuscript. However, the authors did not perform further research or provide further evidence to support this counter-intuitive finding. The authors provided a mixture of expected and unexpected results and a lengthy discussion on which results were consistent with the previous discussion and which are not. But these descriptions and comparisons remain at a superficial level. After finishing reading the manuscript, I still do not understand why cfDNAL929 stimulates the tumor, and why cfDNAB16 inhibits and tumor. The authors need to emphasize and provide more evidence to make this observation robust and trustworthy.

We supplemented our experiments by studying the contribution of mitochondrial DNA to the formation of a pool of various cfDNA, and also checked the presence of TLR9 receptors in B16 melanoma. In accordance with the newly obtained data, we have supplemented and improved the Results and Discussion sections. We have also partially reworked the part regarding changes in gene expression. We have added a Graphic Abstract for future readers. Please see, marked by red.

The current gene selections and presentations in section 2.3 do not fit well with the other 2 result sections. Figure 2 shows a mixture of genes that either change similarly or differentially between cfDNAL929 and cfDNAB16 treated cells. Such result does not clearly support the different cell behavior of cfDNAL929 and cfDNAB16 treated cells in section 2.2 and 2.4. Please revise.

Corrected, please see section 2.3, Figure 2 and Discussion.

Specific comments

Please specify the number of replicates (n=) used in all the figures.

Corrected, please see figure’s legends.

For Table 1 and section 2.1, please present the exact relative abundance value from qPCR. Please provide a plot to visualize the correlation between cfDNA and genomic DNA profile.

We have corrected Table 1 and provided the values in the format as MEAN ± SD.

For line 124-125, what is the statistical test applied and what is the corresponding p value to show statistical insignificant?

In all in vitro experiments, when comparing groups, ANOVA comparison and post-hoc Tukey's test were used. Information was added to Figure 1 legend.

Cell metabolic process is a large term involving thousands of genes. What exactly are the genes in Figure 2C related to? How were these genes selected?

Corrected, please see section 2.3, Figure 2 and Discussion.

Please visualize the comparisons made from line 272-287 with p-values stated and labeled.

We decided to leave the material in Tables 2 and 3 in the form of text material, since in our opinion, in this form this material is more representative. We have added information about p-values.

Round 2

Reviewer 1 Report

Comments and Suggestions for Authors

The manuscript by Filatova et al. provides compelling evidence that sheds light on the potential role of DNA fibroblasts in the tumor microenvironment, specifically in tumor progression as well as the formation of new tumor foci through the transformation of normal cells.

The authors have addressed all the previous comments. Thus, the manuscript can be accepted in its present form.

Reviewer 2 Report

Comments and Suggestions for Authors

The authors made necessary changes to address my concerns. The manuscript may now be published. 

Reviewer 3 Report

Comments and Suggestions for Authors

My comments were properly addressed.

Comments on the Quality of English Language

English writing is good in general.